# Emotion Classification Based on Biophysical Signals and Machine Learning Techniques

**Oana Bălan [1],\* [ID], Gabriela Moise [2] [ID], Livia Petrescu [3] [ID], Alin Moldoveanu [1] [ID], Marius Leordeanu [1] and Florica Moldoveanu [1] [ID]**

[1] Faculty of Automatic Control and Computers, University POLITEHNICA of Bucharest, Bucharest 060042, Romania; alin.moldoveanu@cs.pub.ro (A.M.); marius.leordeanu@cs.pub.ro (M.L.); florica.moldoveanu@cs.pub.ro (F.M.)

[2] Department of Computer Science, Information Technology, Mathematics and Physics (ITIMF), Petroleum-Gas University of Ploiesti, Ploiesti 100680, Romania; gmoise@upg-ploiesti.ro

[3] Faculty of Biology, University of Bucharest, Bucharest 030014, Romania; livia.petrescu@bio.unibuc.ro

\* Correspondence: oana.balan@cs.pub.ro; Tel.: +40722276571

**Abstract:** Emotions constitute an indispensable component of our everyday life. They consist of conscious mental reactions towards objects or situations and are associated with various physiological, behavioral, and cognitive changes. In this paper, we propose a comparative analysis between different machine learning and deep learning techniques, with and without feature selection, for binarily classifying the six basic emotions, namely anger, disgust, fear, joy, sadness, and surprise, into two symmetrical categorical classes (emotion and no emotion), using the physiological recordings and subjective ratings of valence, arousal, and dominance from the DEAP (Dataset for Emotion Analysis using EEG, Physiological and Video Signals) database. The results showed that the maximum classification accuracies for each emotion were: anger: 98.02%, joy:100%, surprise: 96%, disgust: 95%, fear: 90.75%, and sadness: 90.08%. In the case of four emotions (anger, disgust, fear, and sadness), the classification accuracies were higher without feature selection. Our approach to emotion classification has future applicability in the field of affective computing, which includes all the methods used for the automatic assessment of emotions and their applications in healthcare, education, marketing, website personalization, recommender systems, video games, and social media.

**Keywords:** emotion classification; machine learning; feature selection; affective computing; biophysical sensors

## 1. Introduction

Emotions influence our quality of life and how we interact with others. They determine the thoughts we have, the actions we take, subjective perceptions of the world, and our behavioral responses. According to Scherer's theory [1], emotions consist of five synchronized processes, namely cognitive appraisal, bodily symptoms (physiological reactions in the central and autonomic nervous systems), action tendencies (the motivational component that determines us to react or take action), facial or vocal expressions, and feelings (inner experiences, unique for each person apart). Affective computing is the study of systems or devices that can identify, process, and simulate emotions. This domain has applicability in education, medicine, social sciences, entertainment, and so on. The purpose of affective computing is to improve user experience and quality of life, and this is why various emotion models have been proposed over the years and efficient mathematical models applied in order to extract, classify, and analyze emotions.



In the 1970s, Paul Ekman identified six basic emotions [2], namely anger, disgust, fear, joy, sadness, and surprise. Russell and Mehrabian proposed a dimensional approach [3] which states that any emotion is represented relative to three fundamental dimensions, namely valence (positive/pleasurable or negative/unpleasurable), arousal (engaged or not engaged), and dominance (degree of control that a person has over their affective states).

Joy or happiness is a pleasant emotional state, synonymous with contentment, satisfaction and well-being. Sadness is the opposite of happiness, being characterized by grief, disappointment, and distress. Fear emerges in the presence of a stressful or dangerous stimulus perceived by the sensory organs. When the fight or flight response appears, heart rate and respiration rate increase. Also, the muscles become more tense in order to contend with threats in the environment. Anger is defined by fury, frustration, and resentment towards others. Surprise is triggered by an unexpected outcome to a situation, ranging from amazement to shock, whereas disgust is synonymous with dislike, distaste, or repugnance, being the most visceral of all six emotions.

The DEAP database [4] was created with the purpose of developing a music video recommendation system based on the users' emotional responses. The biophysical signals of 32 subjects have been recorded while they were watching 40 one-minute long excerpts of music videos eliciting various emotions. The participants rated each video in terms of valence, arousal, dominance, like/dislike and familiarity on a scale from one to nine. The physiological signals were: galvanic skin response (GSR), plethysmograph (PPG), skin temperature, breathing rate, electromyogram (EMG), and electroencephalography (EEG) from 32 electrodes, decomposed into frequency ranges (theta, slow alpha, alpha, beta, and gamma) and the differences between the spectral power of all symmetrical pairs of electrodes on the left and right brain hemispheres. Other well-known databases are MAHNOB, SEED, and eNTERFACE06_EMOBRAIN. The MAHNOB database [5] contains the physiological signals of 27 subjects in response to 20 stimulus videos who rated arousal, valence, dominance and predictability on a scale from one to nine. The SEED database [6] stores facial videos and EEG data from 15 participants who watched emotional video clips and expressed their affective responses towards them by filling in a questionnaire. The multimodal eNTERFACE06_EMOBRAIN dataset [7] contains EEG, frontal fNIRS and physiological recordings (GSR, respiration rate, and blood volume pressure) from five subjects in response to picture stimuli.

We divided the recordings from de DEAP into six groups, corresponding to the basic six emotions, according to the valence-arousal-dominance ratings. Each emotion has been binary classified into two classes: 1 (emotion) and 0 (lack of emotion). For emotion classification, we have used four deep neural network models and four machine learning techniques. The machine learning techniques were: support vector machine (SVM), linear discriminant analysis (LDA), random forest (RF) and k-nearest neighbors (kNN). These algorithms have been trained and cross-validated, with and without feature selection, on four input sets, containing EEG and peripheral data:

(1) Raw 32-channel EEG values and the peripheral recordings, including hEOG (horizontal electrooculography), vEOG (vertical electrooculography), zEMG (zygomaticus electromyography), tEMG (trapezius electromyography), galvanic skin response (GSR), respiration rate, plethysmography (PPG), temperature;

(2) Petrosian fractal dimensions of the 32 EEG channels and the peripheral recordings mentioned in (1).

(3) Higuchi fractal dimensions of the 32 EEG channels and the peripheral recordings mentioned in (1).

(4) Approximate entropy for each of the 32 EEG channels and the peripheral recordings mentioned in (1).

Feature selection has been ensured by using a Fisher score, principal component analysis (PCA) and sequential forward selection (SFS).

This work is a continuation of the research presented in [8], wherein, using the same techniques, we classified fear by considering it to be of low valence, high arousal, and low dominance. Similarly, classification has been based on the physiological recordings and subjective ratings from the DEAP database. In the current approach, we extend our study of emotion classification by including all six basic emotions from Ekman's theory. Our research has impact in the field of affective computing, as we can understand better the physiological characteristics underlying various emotions. This could lead to the development of effective computational systems that can recognize and process emotional states in the fields of education, healthcare, psychology, and assistive therapy [9,10].

## 2. Materials and Methods

### 2.1. Emotion Models

Various theoretical models of emotions have been developed and most of them have been used for automatic emotion recognition.

Paul Ekman initially considered a set of 6 basic emotions, namely sadness, happiness, disgust, anger, fear and surprise [2]. This model is known as the discrete model of emotions. Later, he expanded the list to 15 emotions: amusement, anger, contempt, contentment, disgust, embarrassment, excitement, fear, guilt, pride in achievement, relief, sadness/distress, satisfaction, sensory pleasure and shame [11]. In 2005, Cohen claimed that empirical evidence does not support the framework of basic emotions and that autonomic responses and pan-cultural facial expressions provide no basis for thinking that there is a set of basic emotions [12].

In contrast to the discrete model, the dimensional model provides ways to express a wide range of emotional states. Using this model, an emotion is described using two or three fundamental features and the affective states are expressed in a multi-dimensional space [13–15]. Russell's circumplex model is an early model, in which an affective state is viewed as a circle in the two-dimensional bipolar space [15]. The proposed dimensions are pleasure and arousal. Pleasure (valence) reflects the positive or negative emotional states, and a value close to zero means a neutral emotion. Arousal expresses the active or passive emotion component. In this space, 28 affective states are represented: happy, delighted, excited, astonished, aroused, tense, alarmed, angry, afraid, annoyed, distressed, frustrated, miserable, sad, gloomy, depressed, bored, droopy, tired, sleepy, calm, relaxed, satisfied, at ease, content, serene, glad, and pleased.

Whissell also used a bi-dimensional space with activation and evaluation as dimensions [14]. Later, he refined his model and proposed the wheel of emotions as follows: quadrant I (positive valence, positive arousal), quadrant II (negative valence, positive arousal), the third quadrant (negative valence, negative arousal) and quadrant IV (positive valence, negative arousal). Examples of emotional states and their positions in the wheel are as follows: joy, happiness, love, surprised, contentment in QI; anger, disgust, fear in QII; sadness, boredom, depression in QIII and relaxation, calm in QIV [16].

Plutchik developed a componential model in which a complex emotion is a mixture of fundamental emotions. The fundamental emotions considered by Plutchik are joy, trust, fear, surprise, sadness, anticipation, anger, and disgust [13].

A three-dimensional model, called the pleasure-arousal-dominance (PAD) model or Valence-Arousal-Dominance (VAD), was introduced by Mehrabian and Russell in [3,17–19]. In the PAD model, there are three independent dimensions: pleasure (valence), which ranges from unhappiness to happiness and expresses the pleasant or unpleasant feeling about something, arousal, the level of affective activation, ranging from sleep to excitement, and dominance, which reflects the level of control of the emotional state, from submissive to dominant. Figure 1 presents the distribution of Ekman's basic emotions within the dimensional emotional space, spanned by the valence, arousal, and dominance axis of the VAD model [20], with ratings taken from Russell and Mehrabian [3].

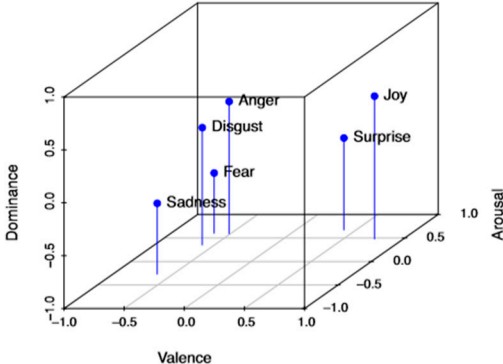

**Figure 1.** The VAD (Valence-Arousal-Dominance) model spanned across the six basic emotions.

Russell and Mehrabian provided in [3] a correspondence between the VAD model and the discrete model of emotions. The values for the six basic emotions, in terms of emotion dimensions, are presented in Table 1.

**Table 1.** Values for the six basic emotions in terms of emotion dimensions.

|  | *Valence* | *Arousal* | *Dominance* |
|---|---|---|---|
| **Anger** | −0.43 | 0.67 | 0.34 |
| **Joy** | 0.76 | 0.48 | 0.35 |
| **Surprise** | 0.4 | 0.67 | −0.13 |
| **Disgust** | −0.6 | 0.35 | 0.11 |
| **Fear** | −0.64 | 0.6 | −0.43 |
| **Sadness** | −0.63 | 0.27 | −0.33 |

## 2.2. Emotions and the Nervous System

In everyday life, each of us is trapped in the chain of emotions, an important component of behavior. Attempts to define and characterize emotions date back to ancient times, but from the 19th century, research has begun to be scientifically documented. It is well known that there is a close correlation between brain functions and emotions. In particular, the limbic system (hypothalamus, thalamus, amygdala, and hippocampus), the paralimbic system, the vegetative nervous system, and the reticular activating system are involved in processing and controlling emotional reactions. Particular importance is given to the prefrontal cortex, anterior cingulate cortex (ACC), nucleus accumbens, and insula.

The limbic system categorizes our emotions into pleasant and unpleasant (valence). Depending on this, chemical neuro-mediators (noradrenaline and serotonin) increase or decrease, influencing the activity of different regions of the body (posture, mimicry, gestures), in response to different emotional states.

The amygdala, a structure that gives an emotional connotation of events and memories, is located deep within the right and left anterior temporal lobes of the brain [21]. The amygdala is a neural switch for fear, anxiety and panic.

The hypothalamus is responsible for processing the incoming signals in response to internal mental events such as pain or anger. Hypothalamus triggers corresponding visceral physiological effects like a raised heart rate, blood pressure, or galvanic skin response [22].

The insula, the part of the limbic system located deep in the lateral sulcus (Sylvius), is part of the primary gustatory cortex. Regarding the perception of emotions, in this region is perceived the feeling of disgust, which comes as a variant of an unpleasant taste. The experience of disgust protects us from the consumption of spoiled or poisonous foods [23,24].

The hippocampus reminds us of the actions responsible for certain emotional states. Hippocampus abnormalities are associated with mood and anxiety disorders [25].

The reticular activating system controls arousal, attention, sleep, wakefulness, and reflexes [26].

### 2.3. The Six Basic Emotions and Their Corresponding Physiological Reactions

Happiness is an emotional state associated with well-being, pleasure, joy, and full satisfaction. This state is characterized by a facial expression in which the mouth corners are raised [27]. Happiness activates the right frontal cortex, the left amygdala, the precuneus and the left insula, involving connections between awareness centers - frontal cortex, the insula and the center of feeling– the amygdala [28].

Sadness, the opposite of happiness and different from depression, is an emotion associated with the feelings of regret, weakness, mental pain and melancholy. This state is characterized by a facial expression that causes lowering the mouth's corners, lifting the inner corner of the upper eyelid, raising and nearing the eyebrows. The angle with the tip upward between the inner corners of the eyebrows is a relevant sign of sadness [27]. At the brain level, sadness is associated with increased activity of the hippocampus, amygdala, right occipital lobe, left insula, and left thalamus [28].

Fear is an innate emotion, considered as an evolutionary mechanism of adaptation to survival, that appears in response to a concrete or anticipated danger. This emotion is controlled by the autonomic nervous system which brings the body into a fight-or-flight state. Fear is characterized by an increasing heart rate and respiratory frequency, peripheral vasoconstriction, perspiration, hyperglycemia, etc. At the brain level, fear activates the bilateral amygdala that communicates with the hypothalamus, the left frontal cortex and other parts of limbic system [28].

Anger is an intense primary emotional state that is part of the fight or flight mechanism, manifested in response to threats or provocations. During the anger state, as a result of the stimulation of the sympathetic vegetative system, a rising of the adrenaline and noradrenaline discharges occurs, followed by an elevation of blood pressure, increasing heart rate and respiratory frequency [27]. Anger activates the right hippocampus, the amygdala, the left and right part of the prefrontal cortex and the insular cortex [28].

Disgust is often associated with avoidance. Unlike other emotions, in the case of disgust the heart rate decreases. At the facial level disgust is characterized by raising the upper lip, wrinkling the nose bridge and raising the cheeks [27]. Disgust implies an activation of the left amygdala, left inferior frontal cortex and insular cortex [28].

Surprise is the hardest emotion to immortalize, being an unexpected and short-lived experience. After the surprise passes, it turns into fear, anger, disgust, or amusement. When someone experiences surprise, the bilateral inferior frontal gyrus and the bilateral hippocampus are activated. The person tends to arch their brows, open the eyes widely and drop their jaw. The hippocampus is also activated, as it is strongly associated with memory and experiences one had or did not have before [28,29].

### 2.4. Biophysical Data

Electroencephalography (EEG) is a method of exploring the electrical potentials of the brain. The encephalogram is the graph obtained from the registration of electric fields at the scalp level. EEG is efficient for detecting affective states, with good temporal resolution. There are four types of waves commonly recorded in humans.

Delta waves have high amplitude and low frequency (0.5–3 Hz). They are characteristic of psychosomatic relaxation states, being recorded in deep sleep phases. They can also be encountered in anesthetic states, following the blocking of nerve transmission through the reticular activating system. They can appear in any cortical region, but predominate in the frontal area [30]. The theta waves have a frequency of 3–8 Hz. This rhythm occurs during low brain activities, sleep, drowsiness or deep meditation. An excess of theta waves is related to artistic, creative, or meditative states. The alpha waves, so-called 'basic rhythm', are oscillations of small amplitude with average frequencies around

8–12 Hz. Under normal conditions, their amplitude increases and decreases regularly, the waves being grouped into characteristic spindle. They appear in the occipital cortex and indicate a normal wakeful state when the human subjects are relaxed or have their eyes closed. The beta waves are characterized by a frequency of 12–30 Hz. Unlike the alpha rhythm, beta waves are highly irregular and signify a desynchronization of cortical neural activity. Their maximum incidence is in the anterior parietal and posterior frontal regions of the brain. This wave is associated with active thinking or concentration and is related to consciousness, brain activities and motor behaviors. The gamma waves are the fastest brainwaves (30–42 Hz), usually found during conscious perception and related to the emotions of happiness and sadness [31]. During memorization tasks, the activation of the gamma waves is visible in the temporal lobe. The predominance of these waves has been associated with the installation of anxiety, stress, or arousal states [32].

Galvanic skin response (GSR) or electrodermal activity (EDA) is a handy and relatively noninvasive tool used to study body reactions to various stimuli, being a successful indicator of physiological and psychological arousal. The autonomic control regulates the internal environment and ensures the body's homeostasis [33]. It is considered that the skin is an organ that responds preponderantly to the action of the sympathetic nervous system through the eccrine sweat gland [34]. For this reason, the data acquisition made from the skin can offer information about the attitude of the body's "fight or flee" reactions. Skin conductance is quantified by applying an electrical potential between two contact points on the skin and measuring the current flow between them. EDA has a background component, namely skin conductance level (SCL), resulting from the interaction between the tonic discharges of the sympathetic innervations and local factors [35], and a fast component - skin conductance responses (SCR), which results from the phasic sympathetic neuronal activity [36]. A high level of SCL indicates a high degree of anxiety [37].

Facial electromyography (EMG) uses the corrugator supercilii ("frowning muscle") activity to track emotional valence.

Heart rate (HR) and HR variability (HRV) are other parameters used to assess human emotions. They have good temporal resolution and can monitor variations or trends of emotions. HRV is associated with cerebral blood flow in the amygdala and in the ventromedial prefrontal cortex [38]. Individuals with high HRV tend to better regulate their emotions [39].

Respiration is an important function for maintaining the homeostasis of the internal environment. Respiratory regulation is achieved by correlating the respiratory centers and the brainstem, the limbic system, and the cerebral cortex. Breathing rate also changes according to emotional responses [40].

*2.5. Machine Learning Techniques for Emotions Classification*

The interest in the field of automatic recognition of emotions is constantly increasing. The data used in emotion recognition systems is primarily extracted from voice, face, text, biophysical signals and body motion [41]. In this section, we performed a brief analysis of the machine learning techniques involved in automatic emotion recognition systems using biophysical data.

Three binary classifications have been performed in [4]: low or high arousal, low or high valence, and low or high liking. The authors used the Gaussian naïve Bayes algorithm for classification, alongside Fisher's linear discriminant for feature selection and leave-one-out cross validation for classification assessment. To measure the performance of the proposed scheme, F1 and average accuracies (ACC) were used. To draw a final conclusion, a decision fusion method was adopted.

Atkinson and Campos [42] used the minimum-redundancy maximum-relevance (mRMR) method for feature selection and Support Vector Machine for binary classification into low/high valence and arousal. The reported accuracy rates were: 73.14% for valence and 73.06% for arousal. The study was performed by extracting and processing the EEG features from the DEAP database. Yoon and Chung [43] used the Pearson correlation coefficient for feature extraction and a probabilistic classifier based on the Bayes theorem for resolving the binary classification problem of low/high valence and arousal discrimination, with an accuracy of about 70% for both: 70.9% for valence and 70.1% for

arousal. For the three-level classification (low/medium/high), the accuracy for high arousal was 55.2% and for valence, 55.4%. Similarly, emotion recognition has been performed based on the EEG data from the DEAP dataset.

A similar approach is presented in Naser and Saha [44], where the SVM algorithm led to accuracies of 66.20%, 64.30%, and 28.90% for classifying arousal, valence, and dominance into low and high groups. Dual-tree complex wavelet packet transform (DT-CWPT) was used for feature selection.

In [45], two classifiers, linear discriminant analysis and SVM were used for two-level classification of valence and arousal. The results showed that SVM produces higher accuracies for arousal and the LDA classifier is better in the case of valence. By applying the SVM technique on the EEG features, classification accuracies of 62.4% and 69.40% were achieved during a music-induced affective state evaluation experiment whereby the users were required to rate their currently perceived emotion in terms of valence and arousal. In the case of the LDA classifiers, the accuracies were 65.6% for valence and 62.4% for arousal. Liu et al [46] conducted two experiments in which visual and audio stimuli were used to evoke emotions. The SVM algorithm, having as input fractal dimension features (FD), statistical and higher order crossings (HOC) extracted from the EEG signals provided the best accuracy for recognizing two emotions - 87.02%, in the case of the audio database and 76.53% in the case of the visual database. The authors provided a comparison between the performances of the proposed strategies applied on their databases and a benchmark database, DEAP. Having DEAP as data source, the mean accuracy for two emotions recognition was 83.73% and 53.7% for recognizing 8 emotions, namely happy, surprised, satisfied, protected, angry, frightened, unconcerned, and sad. A comparative study of four machine learning methods (k-nearest neighbor, SVM, regression tree, Bayesian networks (BNT)) showed that SVM offered the best average accuracy at 85.8%, followed by regression tree with 83.5% for the classification of five types of emotions, namely anxiety, boredom, engagement, frustration, and anger into 3 categories, namely low, medium, and high [47].

In the case of the two-class classification for arousal, valence and like/dislike ratings, for EEG signals, the average accuracy rates were 55.7%, 58.8%, and 49.4% with SVM. Having as input features the peripheral physiological responses, the classification average accuracies recorded 58.9%, 54.2%, and 57.9% [48].

Based on the MAHNOB dataset and using the SVM algorithm with various kernels, Wiem et al. [49] reached a classification accuracy between 57.34% and 68.75% for valence and between 60.83% and 63.63% for arousal when discriminating into low/high groups and between 46.36% and 56.83%, respectively, 50.52% and 54.73% for classification into 3 groups. The features were normalized and a level feature fusion (LFF) algorithm was used. The most relevant features were the electrocardiogram and the respiration volume.

In [50], a deep learning method based on the long-short term memory algorithm was used for classifying low/high valence, arousal and liking based on the EEG raw data from the DEAP dataset [4], with accuracies of 85.45%, 85.65% and 87.99%. Jirayucharoensak et al. [51] used a deep learning network implemented with a stacked autoencoder based on the hierarchical feature learning approach. The input features were the power spectral densities of the EEG signals from the DEAP database, which were selected using the principal component analysis (PCA) algorithm. Covariate Shift Adaptation (CSA) was applied to reduce the non-stationarity in EEG signals. The ratings from 1 to 9 have been divided into 3 levels and mapped into "negative", "neutral", and "positive" for valence and into "passive" "neutral", and "active" for arousal. A leave-one-out cross validation scheme was used to evaluate the performance. They were finally classified with an accuracy of 49.52% for valence and 46.03% for arousal.

A 3D convolutional neural network–based schema has been applied on the DEAP data set in [52] for a two-level classification of valence and arousal. The authors increased the training samples through an augmentation process adding noise signals to the original EEG signals. The schema consisted of 6 layers: input layer, middle layers (two pairs of convolution and max-pooling layers) and a fully-connected output layer. In both convolution layers, rectified linear unit (RELU) is used

as activation function. The recorded accuracies for the proposed method were: 87.44% for valence and 88.49% for arousal. Random forest is not a very common technique used for emotion recognition. In [53], the authors reported a 74% overall accuracy rate for emotions classification into amusement, grief, anger, fear and a baseline state using the random forest classifier. Leave-one-subject-out cross validation was used for evaluating the classifier.

In Table 2, we present the performance of the ML techniques used for emotion recognition.

**Table 2.** Emotions classification performance.

| Reference | Open Data Source | Classifiers | Classification | Feature Selection/Processing | Measure of Performance (%) |
|---|---|---|---|---|---|
| [4] 2012 | DEAP | Gaussian Naïve Bayes | Two-level class: arousal, valence, liking | Fischer's linear discriminant | **F1—scores** 62.9—arousal 65.2—valence 64.2—liking |
| [42] 2016 | DEAP | SVM | Two-level class: arousal, valence | Minimum redundancy Maximum relevance | **Accuracy** 73.06—arousal 73.14—valence |
| [42] 2013 | DEAP | Probabilistic classifier based on Bayes' theorem | Two-level class: arousal, valence | Pearson correlation coefficient | **F1- scores** 74.9—high arousal 62.8—low arousal 74.7—high valence 65.9—low valence **Accuracy** 70.1—high arousal 70.9—high valence |
| [43] 2013 | DEAP | Probabilistic classifier based on Bayes' theorem | Three-level class: arousal, valence | Pearson correlation coefficient | **F1- scores** 63.3 - high arousal 43.3—medium arousal 53.9—low arousal 66.1—high valence 40.9—medium valence 51.8—low valence **Accuracy** 55.2—high arousal 55.4—high valence |
| [44] 2013 | - | SVM | Two-level class | DT-CWPT | **Accuracy** 66.20—arousal 64.30—valence 68.90—dominance |
| [45] 2015 | - | SVM | Two-level class | Stepwise Linear Regression | **Accuracy** (%) 62.4—valence 69.4—arousal |
| [45] 2015 | - | LDA | Two-level class | Stepwise Linear Regression | **Accuracy** (%) 65.6—valence 62.4—arousal |
| [46] 2013 | - | SVM | Discrete emotion (presence or not) | HOC+6 statistical +FD | **Accuracy** (%) Audio database 87.02—2 emotions 76.53—2 emotions Visual database 61.67%—5 emotions 56.6—5 emotions |
| [46] 2013 | DEAP | SVM | Discrete emotion (presence or not) | HOC+6 statistical +FD | **Accuracy** 83.73—2 emotions 53.7—8 emotions |
| [47] 2005 | - | kNN RT BNT SVM | Three-level class | Entire feature set | **Average accuracy** 75.12 83.50 74.03 85.8 |
| [48] 2010 | - | SVM | Two-level class | Fast correlation based filter (FCBF) | **Average accuracy** 54.2—valence 58.9—arousal 57.9—like/dislike |
| [49] 2017 | MAHNOB-HCI | SVM Gaussian kernel | Two-level class | Feature fusion | **Accuracy** 63.63—arousal 68.75—valence |
| [49] 2017 | MAHNOB-HCI | SVM Gaussian kernel | Three-level class | Feature fusion | **Accuracy** 59.57—arousal 57.44—valence |

**Table 2.** *Cont.*

| Reference | Open Data Source | Classifiers | Classification | Feature Selection/Processing | Measure of Performance (%) |
|---|---|---|---|---|---|
| [50] 2017 | DEAP | End-to-end deep learning neural networks | Two-level class | Raw EEG signals | **Average accuracy** 85.65—arousal 85.45—valence 87.99—liking |
| [51] 2014 | DEAP | Deep learning network | Three-level class | PCA PCA CSA CSA | **Accuracy** 50.88—valence 48.64—arousal 53.42—valence 52.03—arousal |
| [52] 2018 | DEAP | 3D Convolutional Neural Networks | Two-level class | Spatiotemporal features are obtained from EEG signals | **F1 score** 86—valence 86—arousal **Accuracy** 87.44—valence 88.49—arousal |
| [53] 2014 | - | RF | Quinary classification: amusement, anger, grief, fear, baseline | Correlation Analysis and t-test | **Correct rate** 25.6—amusement 36.4—anger 74.8—grief 80.1—fear 88.1—baseline |

*2.6. Our Paradigm for Emotions Classification*

In Mehrabian and Russell's model provided in [3], the emotion dimensions (valence, arousal and dominance) are spanned across the interval [–1; 1]. The valence, arousal and dominance ratings from the DEAP database are continuous values in the interval [1; 9]. In order to obtain a correspondence, they have been mapped as follows (Figure 2):

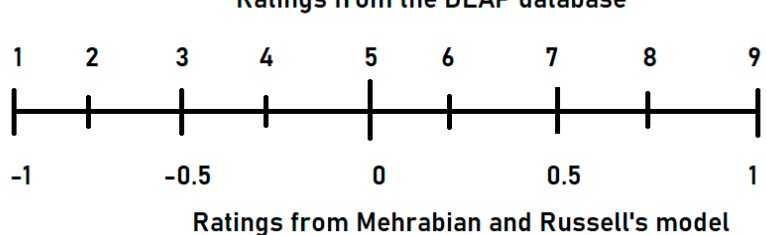

**Figure 2.** Correspondence between the ratings from Mehrabian and Russell's model and the ratings from the DEAP database.

Table 3 presents the intervals of valence, arousal and dominance assigned to each of the six basic emotions, inspired from the values of Mehrabian and Russell's model from Table 1. The ratings of valence and arousal from the DEAP database have been assigned to a larger interval: low ([1;5)) or high ([5;9]). Dominance was the emotion dimension that fluctuated in a smaller interval. Thus, an emotion is characterized by low or high valence/arousal and some degree of dominance spanned across a narrower interval.

Table 4 presents the intervals corresponding to Condition 0, no emotion (or the lack of emotion), and Condition 1, the existence of a certain degree of emotion.

Four input features, sets have been generated after extracting and labelling the data from DEAP: (1) 32-channel raw EEG values and the peripheral recordings: hEOG, vEOG, zEMG, tEMG, GSR, Respiration, PPG, and temperature; (2) Petrosian fractal dimensions of the 32 EEG channels and the peripheral recordings mentioned at (1); (3) Higuchi fractal dimension of the 32 EEG channels and the peripheral recordings mentioned at (1); (4) Approximate entropy of the 32 EEG channels and the peripheral recordings mentioned at (1).

**Table 3.** Valence, arousal, and dominance intervals for the six basic emotions.

| | Valence | | Arousal | | Dominance | |
| --- | --- | --- | --- | --- | --- | --- |
| | **Rating from [3]** | **Rating Adapted from the DEAP Database** | **Rating from [3]** | **Rating Adapted from the DEAP Database** | **Rating from [3]** | **Rating Adapted from the DEAP Database** |
| *Anger* | –0.43 | Low [1; 5) | 0.67 | High [5; 9] | 0.34 | [6;7] |
| *Joy* | 0.76 | High [5;9] | 0.48 | High [5;9] | 0.35 | [6;7] |
| *Surprise* | 0.4 | High [5;9] | 0.67 | High [5;9] | –0.13 | [4;5] |
| *Disgust* | –0.6 | Low [1; 5) | 0.35 | High [5; 9] | 0.11 | [5;6] |
| *Fear* | –0.64 | Low [1; 5) | 0.6 | High [5; 9] | –0.43 | [3;4] |
| *Sadness* | –0.63 | Low [1; 5) | 0.27 | Low [1; 5) | –0.33 | [3;4] |

**Table 4.** Intervals corresponding to Condition 0 and Condition 1.

| | | *Valence* | *Arousal* | *Dominance* |
| --- | --- | --- | --- | --- |
| ***Anger*** | No anger (0) | High [5; 9] | Low [1; 5) | [1;6) or (7;9] |
| | Anger (1) | Low [1; 5) | High [5; 9] | [6;7] |
| *Joy* | No joy (0) | Low [1; 5) | Low [1; 5) | [1;6) or (7;9] |
| | Joy (1) | High [5; 9] | High [5; 9] | [6;7] |
| *Surprise* | No surprise (0) | Low [1; 5) | Low [1; 5) | [1;4) or (5;9] |
| | Surprise (1) | High [5; 9] | High [5; 9] | [4;5] |
| *Disgust* | No disgust (0) | High [5; 9] | Low [1; 5) | [1;5) or (6;9] |
| | Disgust (1) | Low [1; 5) | High [5; 9] | [5;6] |
| *Fear* | No fear (0) | High [5; 9] | Low [1; 5) | [1;3) or (4;9] |
| | Fear (1) | Low [1; 5) | High [5; 9] | [3;4] |
| *Sadness* | No sadness (0) | High [5; 9] | High [5; 9] | [1;3) or (4;9] |
| | Sadness (1) | Low [1; 5) | Low [1; 5) | [3;4] |

The DEAP database contains 40 valence/arousal/dominance ratings for each of the 32 subjects. For the emotion of anger, there were 28 ratings in the database for Condition 1—Anger and 239 ratings for the Condition 0—No anger. In order to have a balanced distribution of responses for classification, we used 28 ratings for Condition 1 and 28 ratings for Condition 0, so we took the minimum between both. Every physiological recording had a duration of 60 s. Thus, in order to obtain a larger training database, we have divided the 60 s long recordings into 12 segments, each being 5 s long. Thus, for anger we obtained a training dataset of 672 entries that was fed to the classification algorithms. Table 5 presents, for each emotion, the number of entries for Conditions 0 and 1 and the total number of 5-s long segments that have been fed as input data to the classification algorithms.

For binary classifying the emotion ratings into Condition 1 (emotion) and Condition 0 (lack of emotion), we applied four machine and deep learning algorithms, with and without feature selection, similarly to the experiment described in [8], where we classified the emotion of fear. Our input features were: EEG (raw values/approximate entropy/Petrosian fractal dimension/Higuchi fractal dimension) and peripheral signals, hEOG, vEOG, zEMG, tEMG, GSR, respiration rate, PPG and temperature. We constructed models based on four deep neural networks (DNN1-DNN4) with various numbers of

hidden layers and neurons per layer. The machine learning techniques employed were SVM, RF, LDA and kNN. As feature selection algorithms, we used Fisher selection, PCA and SFS.

**Table 5.** Number of entries for each emotion.

|  | Number of entries Condition 1 | Number of entries Condition 0 | Total number of entries (5 s long) |
|---|---|---|---|
| *Anger* | Anger 28 | No anger 239 | 672 |
| *Joy* | Joy 117 | No joy 249 | 2808 |
| *Surprise* | Surprise 201 | No surprise 233 | 4824 |
| *Disgust* | Disgust 61 | No disgust 186 | 1464 |
| *Fear* | Fear 81 | No fear 160 | 1944 |
| *Sadness* | Sadness 89 | No sadness 337 | 2136 |

Higuchi fractal dimension (HFD) is a non-linear method highly used in the analysis of biological signals. It originates from chaos theory and has been used for 30 years as a modality of measuring signals dynamics and complexity. It has been used for detecting hidden information contained in biophysical time series with the help of fractals, which, despite scaling, preserve the structure and shape of complex signals. There are many methods for calculating fractal dimensions, such as Katz's, Petrosian's or Higuchi's [54–56]. Approximate Entropy (ApEn) is a measure of regularity in the time domain which determines the predictability of a signal by comparing the number of matching sequences of a given length with the number of matching sequences one increment longer [57]. In regular data series, knowing the previous values enables the prediction of the subsequent ones. A high value of ApEn is associated with random and unpredictable variation, while a low value correlates with regularity and predictability in a time series [58].

DNN1 has one input layer, three hidden layers with 300 neurons per layer, and one output layer. The input layer contains 40 neurons, corresponding to the 32 EEG data (raw values/Petrosian fractal dimensions/Higuchi fractal dimensions/approximate entropy) and 8 peripheral data (hEOG, vEOG, zEMG, tEMG, GSR, respiration rate, PPG and temperature). Petrosian fractal dimensions, Higuchi fractal dimensions and approximate entropy have been computed using the functions from the PyEEG library [59]. The output layer generates two possible results: 0 or 1. In the output layer, we used the binary crossentropy loss function and sigmoid activation function. Also, the model uses the Adam gradient descent optimization algorithm and the rectified linear unit (RELU) activation function on each layer. The network is organized as a multi-layer perceptron network. The input data has been standardized to zero mean and unit variance. The Keras classifier [60] had 1000 epochs for training and a batch size of 20. Cross-validation has been performed by using the k-fold method with k = 10 splits and the leave-one-out method, which takes each sample as test set and keeps the remaining samples in the training set. However, the leave-one-out method is more computationally demanding than k-fold. The model has been trained and cross-validated for 10 times and we calculated the average accuracy and F1 score across these 10 iterations. Each time, the input data has been shuffled before being divided into the training and test datasets.

DNN2 has 3 hidden layers and 150 neurons/layer, DNN3 has 6 hidden layers with 300 neurons/layer, and DNN4 has 6 hidden layers with 150 neurons/layer. Their configuration and method of training and cross-validating is similar to DNN1. Feature selection was not necessary for the DNNs, as the dropout regularization technique prevents overfitting.

For the SVM method, we used the radial basis function kernel (rbf). For Random Forest, the number of trees in the forest has been set to 10 (default value for the n_estimators parameter in the RandomForestClassifier method from the scikit learn library [61]). The function that measures the

quality of the split has been "entropy", that divides based on information gain. For kNN, the number of neighbors has been set to 7. For SVM, LDA, RF, and KNN, the input data has been divided into 70% training and 30% test using the train_test_split method from the scikit learn library. This function makes sure that each time, the data is shuffled before dividing into the training and test datasets. The input data has been also standardized in order to reduce it to zero mean and unit variance.

These classification methods have been trained and cross-validated 10 times, without feature selection and with the Fisher, PCA, and SFS feature selection methods. In a similar way to the DNNs, we calculated the average accuracy and F1 score across these 10 iterations. The Fisher score has been calculated on the training dataset and then the first, most relevant 20 features have been selected. Consequently, a machine learning model (SVM/RF/LDA/kNN) has been constructed and cross-validated based on these relevant features. The PCA algorithm retains 99% of the data variance (the n_components parameter of the PCA method from scikit learn has been set to 0.99). The SFS classifier selects the best feature combination containing between 3 and 20 features.

## 3. Results

The cross-validation results obtained after training and testing on the data using the machine and deep learning methods, with k-fold cross validation, for each basic emotion, are presented in Tables 6–11. The numbers written in bold correspond to the maximum F1 scores and accuracies. Table 12 presents the most important features for each of the six basic emotions, based on the Fisher score and SFS algorithm. The accuracies obtained using the leave-one-out method for cross-validation are with 5%–10% lower, but the hierarchy of results is preserved, not affecting the classification ranking.

Figure 3 presents the decision tree obtained for classifying anger using RF with raw EEG data and peripheral features, without feature selection.

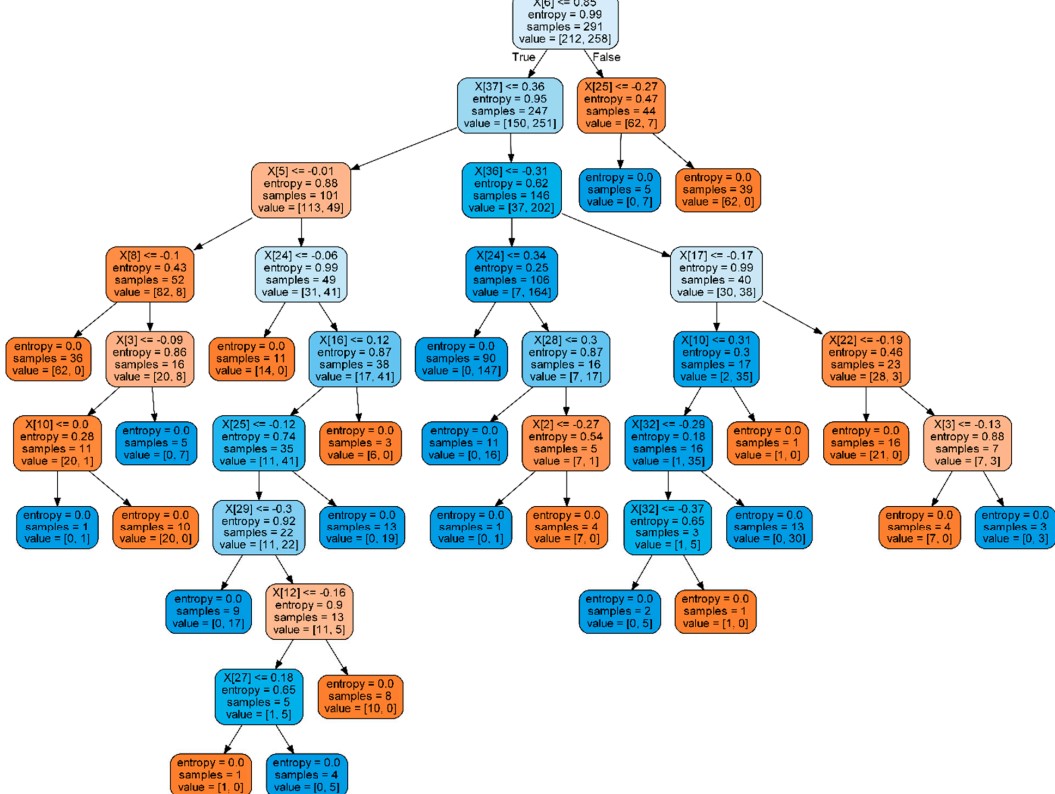

**Figure 3.** Decision tree for Anger using RF with raw EEG data and peripheral features, without feature selection.

**Table 6.** Classification F1 scores and accuracies for anger. (The numbers written in bold correspond to the maximum F1 scores and accuracies.)

| Type of Feature Selection | Classifier | Anger | | | | | | | |
|---|---|---|---|---|---|---|---|---|---|
| | | Raw | | Petrosian | | Higuchi Fractal Dimension | | Approximate Entropy | |
| | | F1 Score (%) | Accuracy (%) | F1 Score (%) | Accuracy (%) | F1 Score (%) | Accuracy (%) | F1 Score (%) | Accuracy (%) |
| No feature selection | DNN1 | 91.22 | 91.22 | 90.03 | 90.03 | 90.03 | 90.03 | 74.46 | 74.55 |
| | DNN2 | 87.80 | 87.76 | 87.05 | 87.04 | 81.24 | 81.25 | 73.06 | 73.07 |
| | DNN3 | 93.30 | 93.30 | 87.50 | 87.50 | 89.73 | 89.73 | 75.15 | 75.15 |
| | DNN4 | 88.39 | 88.39 | 84.97 | 84.96 | 80.08 | 80.21 | 71.20 | 71.28 |
| | SVM | 92.57 | 92.58 | **98.02** | **98.02** | 98.02 | 98.02 | 68.28 | 68.32 |
| | RF | **96.04** | **96.04** | 95.05 | 95.04 | 97.52 | 97.52 | **92.55** | **92.57** |
| | LDA | 85.64 | 85.63 | 92.08 | 92.08 | 96.04 | 96.04 | 68.81 | 68.81 |
| | kNN | 93.56 | 93.53 | 95.05 | 95.04 | 97.03 | 97.03 | 85.15 | 85.15 |
| Fisher | SVM | 86.14 | 86.04 | **95.05** | **95.05** | 94.05 | 94.06 | 69.70 | 69.80 |
| | RF | 95.54 | 95.54 | 92.57 | 92.56 | 88.15 | 88.12 | **92.08** | **92.08** |
| | LDA | 80.69 | 80.64 | 93.07 | 93.07 | 87.12 | 87.13 | 61.74 | 62.38 |
| | kNN | **97.52** | **97.52** | 93.56 | 93.55 | 93.56 | 93.56 | 88.09 | 88.12 |
| PCA | SVM | 93.42 | 93.42 | **97.67** | **97.67** | 98.32 | 98.32 | 81.93 | 82.08 |
| | RF | 92.43 | 92.42 | 92.28 | 92.26 | 93.62 | 93.61 | 86.42 | 86.44 |
| | LDA | 81.24 | 81.15 | 91.73 | 91.73 | 94.75 | 94.75 | 65.93 | 66.09 |
| | kNN | **93.96** | **93.95** | 95.05 | 95.05 | 95.59 | 95.59 | **87.37** | **87.38** |
| SFS | SVM | **91** | **91** | **91** | **91** | 84 | 84 | 71 | 71 |
| | RF | 86 | 86 | 86 | 86 | 83 | 83 | 78 | 78 |
| | LDA | **91** | **91** | **91** | **91** | **85** | **85** | 66 | 66 |
| | kNN | **91** | **91** | **91** | **91** | 83 | 83 | **79** | **79** |

**Table 7.** Classification F1 scores and accuracies for Joy. (The numbers written in bold correspond to the maximum F1 scores and accuracies.)

| Type of Feature Selection | Classifier | Joy | | | | | | | |
|---|---|---|---|---|---|---|---|---|---|
| | | Raw | | Petrosian | | Higuchi Fractal Dimension | | Approximate Entropy | |
| | | F1 Score (%) | Accuracy (%) | F1 Score (%) | Accuracy (%) | F1 Score (%) | Accuracy (%) | F1 Score (%) | Accuracy (%) |
| No feature selection | DNN1 | 82.29 | 82.30 | 80.62 | 80.63 | 79.46 | 79.49 | 69.96 | 70.16 |
| | DNN2 | 80.30 | 80.34 | 78.10 | 78.10 | 76.14 | 76.18 | 67.24 | 67.38 |
| | DNN3 | 83.62 | 83.65 | 80.91 | 80.91 | 80.51 | 80.52 | 72.14 | 72.15 |
| | DNN4 | 81.58 | 81.59 | 79.02 | 79.02 | 76.80 | 76.85 | 67.17 | 67.45 |
| | SVM | 83.60 | 83.63 | 86.47 | 86.48 | 84.45 | 84.46 | 65.15 | 65.95 |
| | RF | 90.25 | 90.27 | 86.31 | 86.36 | 87.57 | 87.66 | **86.40** | **86.48** |
| | LDA | 71.63 | 71.65 | 70.94 | 70.94 | 72.12 | 72.12 | 65.02 | 65.12 |
| | kNN | **91.22** | **91.22** | **87.90** | **87.90** | **87.60** | **87.60** | 83.35 | 83.39 |
| Fisher | SVM | 78.48 | 78.65 | 83.98 | 83.99 | 79.58 | 79.60 | 68.65 | 69.16 |
| | RF | 89.55 | 89.56 | 80.64 | 80.78 | 81.09 | 81.14 | **80.37** | **80.43** |
| | LDA | 64.03 | 64.29 | 68.82 | 68.92 | 67.03 | 67.02 | 64.44 | 64.53 |
| | kNN | **89.92** | **89.92** | **85.76** | **85.77** | **83.75** | **83.75** | 74.01 | 74.02 |
| PCA | SVM | 83.17 | 83.21 | 86.39 | 86.39 | 84.95 | 84.96 | 72.01 | 72.41 |
| | RF | 88.48 | 88.51 | 81.83 | 81.91 | 84.91 | 84.95 | 82.20 | 82.20 |
| | LDA | 70.62 | 70.77 | 71.71 | 71.71 | 72.32 | 72.33 | 63.54 | 63.74 |
| | kNN | **89.83** | **89.83** | **88.08** | **88.08** | **87.55** | **87.56** | 82.25 | 82.27 |
| SFS | SVM | 98 | 98 | 67 | 67 | 67 | 67 | 66 | 66 |
| | RF | 96 | 96 | **70** | **70** | 70 | 70 | **70** | **70** |
| | LDA | **100** | **100** | 65 | 65 | 65 | 65 | 61 | 61 |
| | kNN | 98 | 98 | 66 | 66 | 66 | 66 | 66 | 66 |

**Table 8.** Classification F1 scores and accuracies for surprise. (The numbers written in bold correspond to the maximum F1 scores and accuracies.)

| Type of Feature Selection | Classifier | Surprise | | | | | | | |
|---|---|---|---|---|---|---|---|---|---|
| | | Raw | | Petrosian | | Higuchi Fractal Dimension | | Approximate Entropy | |
| | | F1 Score (%) | Accuracy (%) | F1 Score (%) | Accuracy (%) | F1 Score (%) | Accuracy (%) | F1 Score (%) | Accuracy (%) |
| No feature selection | DNN1 | 70.89 | 70.94 | 78.04 | 78.05 | 76.41 | 76.43 | 71.50 | 71.54 |
| | DNN2 | 69.74 | 69.88 | 74.94 | 74.94 | 74.19 | 74.23 | 69.55 | 69.57 |
| | DNN3 | 71.41 | 71.41 | 77.67 | 77.67 | 78.05 | 78.11 | 70.47 | 71.02 |
| | DNN4 | 68.92 | 68.93 | 76.12 | 76.12 | 74.42 | 74.42 | 69.61 | 69.94 |
| | SVM | 71.52 | 71.75 | 80.92 | 80.94 | 81.84 | 81.84 | 63.25 | 65.06 |
| | RF | 83.91 | 83.98 | 82.12 | 82.25 | 80.51 | 80.80 | 81.22 | 81.49 |
| | LDA | 59.79 | 59.88 | 63.73 | 63.74 | 67.73 | 67.75 | 59.74 | 59.74 |
| | kNN | **85.01** | **85.01** | **84.74** | **84.74** | **83.64** | **83.63** | **81.30** | **81.30** |
| Fisher | SVM | 70.37 | 70.86 | 75.76 | 75.76 | 72.50 | 72.51 | 62.21 | 63.54 |
| | RF | **81.85** | **81.91** | 76.97 | 77.07 | **80.69** | **80.80** | **82.59** | **82.73** |
| | LDA | 62.52 | 62.71 | 58.48 | 58.49 | 60.43 | 60.43 | 59.58 | 59.60 |
| | kNN | 80.94 | 80.94 | **80.59** | **80.59** | 79.14 | 79.14 | 79.42 | 79.42 |
| PCA | SVM | 73.57 | 73.71 | 80.74 | 80.74 | 80.20 | 80.20 | 70.46 | 70.50 |
| | RF | 81.34 | 81.40 | 79.21 | 79.29 | 81.47 | 81.52 | 78.36 | 78.42 |
| | LDA | 60.65 | 60.71 | 65.60 | 65.62 | 66.81 | 66.84 | 60.02 | 60.12 |
| | kNN | **83.60** | **83.60** | **84.81** | **84.81** | **82.94** | **82.94** | **79.71** | **79.72** |
| SFS | SVM | **96** | **96** | 61 | 61 | 61 | 61 | 61 | 61 |
| | RF | 90 | 90 | **66** | **66** | **64** | **64** | **65** | **65** |
| | LDA | 93 | 93 | 58 | 58 | 58 | 58 | 61 | 61 |
| | kNN | 92 | 92 | 62 | 62 | 62 | 62 | 63 | 63 |

**Table 9.** Classification F1 scores and accuracies for Disgust. (The numbers written in bold correspond to the maximum F1 scores and accuracies.).

| Type of Feature Selection | Classifier | Disgust | | | | | | | |
|---|---|---|---|---|---|---|---|---|---|
| | | Raw | | Petrosian | | Higouchi Fractal Dimension | | Approximate Entropy | |
| | | F1 Score (%) | Accuracy (%) | F1 Score (%) | Accuracy (%) | F1 Score (%) | Accuracy (%) | F1 Score (%) | Accuracy (%) |
| No feature selection | DNN1 | 84.65 | 84.70 | 85.04 | 85.04 | 87.08 | 87.09 | 68.90 | 68.99 |
| | DNN2 | 80.80 | 80.81 | 84.02 | 84.02 | 82.79 | 82.79 | 65.38 | 65.71 |
| | DNN3 | 87.07 | 87.09 | 85.92 | 85.93 | 87.70 | 87.70 | 67.84 | 68.44 |
| | DNN4 | 82.57 | 82.65 | 83.54 | 83.54 | 81.56 | 81.56 | 65.96 | 66.80 |
| | SVM | 86.82 | 86.82 | 91.13 | 91.14 | **91.59** | **91.59** | 64.27 | 65 |
| | RF | **93.63** | **93.64** | 91.36 | 91.36 | 90.19 | 90.23 | **83.14** | **83.18** |
| | LDA | 74.72 | 74.77 | 84.32 | 84.32 | 85.91 | 85.91 | 58.72 | 59.09 |
| | kNN | 92.03 | 92.05 | **95** | **95** | 91.36 | 91.36 | 82.25 | 82.27 |
| Fisher | SVM | 83.20 | 83.18 | **90** | **90** | 90.22 | 90.23 | 64.77 | 65.45 |
| | RF | 89.32 | 89.32 | 84.52 | 84.55 | **90.23** | **90.23** | **75.26** | **75.23** |
| | LDA | 72.27 | 72.27 | 80.45 | 80.45 | 83.39 | 83.41 | 61.97 | 62.73 |
| | kNN | **89.74** | **89.77** | 88.64 | 88.64 | 89.31 | 89.32 | 66.60 | 66.59 |
| PCA | SVM | 86.40 | 86.41 | 92.93 | 92.93 | **93.55** | **93.55** | 74.77 | 74.95 |
| | RF | 87.42 | 87.43 | 87.64 | 87.66 | 90.11 | 90.11 | 78.74 | 78.80 |
| | LDA | 72.37 | 72.43 | 85.28 | 85.30 | 87.52 | 87.52 | 61.61 | 62 |
| | kNN | **90.84** | **90.84** | **93.89** | **93.89** | 92.43 | 92.43 | **82.05** | **82.05** |
| SFS | SVM | 72 | 72 | 73 | 73 | **75** | **75** | **64** | **64** |
| | RF | 65 | 65 | 65 | 65 | 66 | 66 | 62 | 62 |
| | LDA | **76** | **76** | **77** | **77** | 74 | 74 | 62 | 62 |
| | kNN | 74 | 74 | 73 | 73 | 67 | 67 | 57 | 57 |

**Table 10.** Classification F1 scores and accuracies for fear. (The numbers written in bold correspond to the maximum F1 scores and accuracies.).

| Type of Feature Selection | Classifier | Fear | | | | | | | |
|---|---|---|---|---|---|---|---|---|---|
| | | Raw | | Petrosian | | Higuchi Fractal Dimension | | Approximate Entropy | |
| | | F1 Score (%) | Accuracy (%) | F1 Score (%) | Accuracy (%) | F1 Score (%) | Accuracy (%) | F1 Score (%) | Accuracy (%) |
| No feature selection | DNN1 | 82.86 | 82.87 | 78.54 | 78.55 | 81.22 | 81.22 | 66.96 | 67.03 |
| | DNN2 | 79.45 | 79.53 | 75.31 | 75.31 | 79.83 | 79.84 | 63.69 | 63.84 |
| | DNN3 | 84.88 | 84.93 | 78.61 | 78.65 | 80.97 | 81.02 | 67.25 | 67.34 |
| | DNN4 | 82.33 | 82.46 | 75.87 | 75.87 | 78.65 | 78.65 | 63.26 | 63.32 |
| | SVM | 80.21 | 80.48 | 86.82 | 86.82 | 87.15 | 87.16 | 66.02 | 66.95 |
| | RF | 89.52 | 89.55 | 88.20 | 88.18 | 84.41 | 84.42 | 79.26 | 79.28 |
| | LDA | 68.64 | 68.66 | 70.93 | 71.23 | 77.86 | 77.91 | 57.15 | 57.36 |
| | kNN | **90.75** | **90.75** | **89.72** | **89.73** | **89.04** | **89.04** | **80.66** | **80.65** |
| Fisher | SVM | 74.37 | 74.49 | 78.50 | 78.60 | 82.36 | 82.36 | 67.72 | 68.84 |
| | RF | **88.85** | **88.87** | 78.18 | 78.25 | 80.39 | 80.48 | 79.27 | 79.28 |
| | LDA | 65.24 | 65.24 | 69.39 | 69.52 | 72.43 | 72.43 | 59.32 | 59.76 |
| | kNN | 86.98 | 86.99 | **80.82** | **80.82** | **83.39** | **83.39** | **79.42** | **79.45** |
| PCA | SVM | 80.53 | 80.77 | 87.25 | 87.26 | **89.77** | **89.78** | 72.73 | 73.39 |
| | RF | 86.98 | 87.02 | 82.71 | 82.77 | 86.74 | 86.76 | 76.75 | 76.78 |
| | LDA | 62.09 | 62.14 | 73.18 | 73.20 | 77.19 | 77.19 | 57.62 | 57.69 |
| | kNN | **89.21** | **89.23** | **89.95** | **89.95** | 89.38 | 89.38 | **82.25** | **82.26** |
| SFS | SVM | 65 | 65 | 66 | 66 | 65 | 65 | 60 | 60 |
| | RF | 61 | 61 | 61 | 61 | 62 | 62 | **61** | **61** |
| | LDA | **69** | **69** | **69** | **69** | **73** | **73** | 59 | 59 |
| | kNN | 61 | 61 | 61 | 61 | 65 | 65 | 59 | 59 |

**Table 11.** Classification F1 scores and accuracies for sadness. (The numbers written in bold correspond to the maximum F1 scores and accuracies.).

| Type of Feature Selection | Classifier | Sadness | | | | | | | |
|---|---|---|---|---|---|---|---|---|---|
| | | Raw | | Petrosian | | Higuchi Fractal Dimension | | Approximate Entropy | |
| | | F1 Score (%) | Accuracy (% | F1 Score (%) | Accuracy (%) | F1 Score (%) | Accuracy (%) | F1 Score (%) | Accuracy (%) |
| No feature selection | DNN1 | 80.17 | 80.20 | 81.79 | 81.79 | 83.70 | 83.71 | 68.11 | 68.12 |
| | DNN2 | 78.52 | 78.56 | 79.07 | 79.07 | 82.49 | 82.49 | 67.39 | 67.46 |
| | DNN3 | 81.72 | 81.74 | 81.95 | 81.98 | 83.31 | 83.33 | 69.71 | 69.76 |
| | DNN4 | 79.56 | 79.59 | 79.19 | 79.21 | 83.13 | 83.15 | 65.90 | 66.10 |
| | SVM | 76.26 | 76.91 | 86.90 | 86.90 | **90.80** | **90.80** | 65.21 | 65.68 |
| | RF | **87.49** | **87.52** | 84.18 | 84.24 | 84.52 | 84.56 | **81.86** | **81.90** |
| | LDA | 69.37 | 69.42 | 75.82 | 75.82 | 82.37 | 82.37 | 47.12 | 51.17 |
| | kNN | 86.81 | 86.90 | **90.17** | **90.17** | 89.06 | 89.08 | 80.50 | 80.50 |
| Fisher | SVM | 73.96 | 74.26 | 80.97 | 80.97 | **86.43** | **86.43** | 59 | 60.22 |
| | RF | 84.24 | 84.24 | 81.37 | 81.44 | 84.83 | 84.87 | 64.43 | 64.43 |
| | LDA | 66.07 | 66.61 | 69.61 | 69.58 | 78.78 | 78.78 | 50.08 | 50.08 |
| | kNN | **85.76** | **85.80** | 83.29 | 83.31 | 84.71 | 84.71 | **76.45** | **76.44** |
| PCA | SVM | 79.63 | 79.92 | 87.21 | 87.21 | **89.31** | **89.31** | 74.56 | 74.85 |
| | RF | 85.55 | 85.62 | 82.38 | 82.45 | 86.49 | 86.52 | 80.11 | 80.14 |
| | LDA | 58.91 | 58.99 | 74.47 | 74.46 | 80.31 | 80.31 | 52.36 | 53.29 |
| | kNN | **88.14** | **88.17** | **88.53** | **88.53** | 88.12 | 88.13 | **82.56** | **82.56** |
| SFS | SVM | 63 | 63 | **68** | **68** | 69 | 69 | 63 | 63 |
| | RF | **65** | **65** | 66 | 66 | 66 | 66 | 58 | 58 |
| | LDA | **65** | **65** | 67 | 67 | **71** | **71** | **64** | **64** |
| | kNN | 61 | 61 | 63 | 63 | 61 | 61 | 58 | 58 |

**Table 12.** Important features for each emotion.

|  | *Raw* | *Petrosian* | *Higuchi* | *Approximate Entropy* |
|---|---|---|---|---|
| *Anger* | tEMG | F3 | FC1 | tEMG |
|  | Respiration | AF3 | AF3 | Respiration |
|  | O2 | tEMG | F3 | GSR |
|  | P3 | Respiration | CP5 | PPG |
|  | C3 | FC1 | tEMG | vEOG |
| *Joy* | GSR | GSR | Cz | GSR |
|  | FC1 | Oz | GSR | Respiration |
|  | PO3 | zEMG | P8 | zEMG |
|  | C3 | O1 | P3 | hEOG |
|  | Cz | PO3 | T7 | vEOG |
| *Surprise* | GSR | GSR | GSR | GSR |
|  | Cz | FC1 | FC1 | PPG |
|  | C3 | FC2 | Cz | vEOG |
|  | Oz | Cz | P3 | Respiration |
|  | C4 | CP2 | Pz | zEMG |
| *Disgust* | vEOG | FC2 | vEOG | vEOG |
|  | FC5 | vEOG | T7 | hEOG |
|  | C3 | Oz | AF3 | GSR |
|  | P7 | PO3 | hEOG | CP5 |
|  | Respiration | FP1 | CP5 | Oz |
| *Fear* | tEMG | FC1 | FC1 | vEOG |
|  | hEOG | F4 | F4 | zEMG |
|  | vEOG | T8 | FC2 | Respiration |
|  | zEMG | Cz | AF4 | hEOG |
|  | Cz | FC2 | Pz | GSR |
| *Sadness* | CP1 | FC1 | FC1 | PPG |
|  | F8 | FP1 | P3 | Temperature |
|  | P7 | AF3 | O1 | tEMG |
|  | Cz | FC2 | FP1 | Oz |
|  | Respiration | Oz | AF3 | zEMG |

## 4. Discussion

Table 13 presents the best classification F1 scores for each emotion, with and without feature selection. Without feature selection, kNN has been selected in 13 cases, followed by Random Forest (seven times) and SVM (four times). For anger, the highest classification accuracy has been obtained for Petrosian and Higuchi fractal dimension, using SVM (98.02%). For joy, the highest classification accuracy has been achieved by kNN using Petrosian values (87.9%). For surprise, kNN with raw EEG values (85.01%), disgust—kNN with Petrosian values (95%), fear—kNN with raw EEG values (90.75%), sadness—SVM with Higuchi fractal dimensions (90.8%).

With feature selection, kNN has been selected in 12 cases, random forest seven times, SVM five times and LDA one time. SFS has been selected two times and Fisher score 14 times. For anger, the highest classification accuracy has been obtained for raw data using kNN and Fisher (97.52%). For Joy,

the highest classification accuracy has been achieved by LDA and SFS using raw values (100%). For surprise, SVM and SFS with raw EEG values (96%), disgust, random forest and Fisher with Higuchi fractal dimensions (90.23%), fear, kNN and Fisher with Higuchi fractal dimensions (83.39%), and sadness, SVM and Fisher with Higuchi fractal dimensions (86.43%).

**Table 13.** Best classification F1 scores for each emotion.

| | Raw | | Petrosian | | Higuchi Fractal Dimension | | Approximate Entropy | |
|---|---|---|---|---|---|---|---|---|
| | No Feature Selection | With Feature Selection | No Feature Selection | With Feature Selection | No Feature Selection | With Feature Selection | No Feature Selection | With Feature Selection |
| Anger | Random Forest 96.04% | kNN Fisher 97.52% | SVM 98.02% | SVM Fisher 95.05% | SVM 98.02% | SVM Fisher 94.05% | Random Forest 92.55% | Random Forest Fisher 92.08% |
| Joy | kNN 91.22% | LDA SFS 100% | kNN 87.9% | kNN Fisher 85.76% | kNN 87.60% | kNN Fisher 83.75% | Random Forest 86.40% | Random Forest Fisher 80.37% |
| Surprise | kNN 85.01% | SVM SFS 96% | kNN 84.75% | kNN Fisher 80.59% | kNN 83.64% | Random Forest Fisher 80.69% | kNN 81.30% | kNN Fisher 82.59% |
| Disgust | Random Forest 93.63% | kNN Fisher 89.74% | kNN 95% | SVM Fisher 90% | SVM 91.59% | Random Forest Fisher 90.23% | Random Forest 83.14% | Random Forest Fisher 75.26% |
| Fear | kNN 90.75% | Random Forest Fisher 80.85% | kNN 89.72% | kNN Fisher 80.82% | kNN 89.04% | kNN Fisher 83.39% | kNN 80.66% | kNN Fisher 79.45% |
| Sadness | Random Forest 87.49% | kNN Fisher 85.76% | kNN 90.17% | kNN Fisher 83.29% | SVM 90.8% | SVM Fisher 86.43% | Random Forest 81.86% | kNN Fisher 76.45% |

For anger, disgust, fear and sadness, the classification accuracies have been higher without feature selection. The SFS feature selection algorithm lead to higher accuracies for joy and surprise.

According to Table 12, the most important features for anger were tEMG and respiration. This result is consistent with reality, because in conditions of anger, anxiety and stress, besides intensifying the breathing, there is also an accumulation of tension in the muscles located between the forehead and the shoulders (tension triangles). Thus, corrugator muscles are responsible for forehead frowning, the masseter and the temporalis muscles are responsible for jaw clenches, while the trapezius muscles are responsible for the neck tightening and the shoulders rising.

The most important features for joy were GSR and zEMG. Dynamic facial expressions of joy determine an intense activity of the zygomatic muscle, which pulls up and laterally the corners of the lips to sketch a smile. High skin conductance entropy indicates body arousal.

The most important features for surprise are GSR and FC1. Although surprise is an emotion with neutral valence, it is frequently associated with increased GSR.

According to existing studies, disgust suppresses attention, in order to minimize the visual contact with the threatening agent. This explains the movement of the eyeballs horizontally and vertically (vEOG and hEOG), which are the most important features for disgust (Table 12).

Fear is characterized by opening the eyes and rotating the eyeballs horizontally and vertically, for danger identification (vEOG, hEOG), stretching the mouth (zEMG), and opening the nostrils for better tissue oxygenation. Activation of the frontal cortex (FC1, F4) aims to stimulate motor areas and prepare the body for escape or fight.

Sadness involves an increasing activity, mostly in the left prefrontal cortex and in the structures of the limbic system, as we can see from our most selected traits: FC1 and FP1 (Table 12).

The maximum accuracies obtained for classifying the six basic emotions into two classes (1—emotion and 0—no emotion) were higher than those obtained for classifying into low/high valence/arousal—62%/56% [4], 73% [42], 70% [43], 85% using the long-short term memory algorithm

(all using the data from the DEAP database), 66%/64% [44], 62%/69% [45], 55%/58% [62], and 68%/54% using the data from the MAHNOB database [49].

Liu [46] achieved a classification accuracy of 53% for recognizing eight emotions using Fractal Dimension Features with SVM, while we obtained accuracies of over 83% using Higuchi Fractal Dimensions and kNN. Our results are comparable to those obtained by Liu [47] who reached accuracies of 85% with SVM and 83% using a regression tree for classifying anxiety, boredom, engagement, frustration, and anger into three categories, namely low, medium, and high.

## 5. Conclusions

This paper presented a comparative analysis between various machine learning and deep learning techniques for classifying the six basic emotions from Ekman's model [2], namely anger, disgust, fear, joy, sadness, and surprise, using physiological recordings and the valence/arousal/dominance ratings from the DEAP database. DEAP is the most well-known and exhaustive multimodal dataset for analyzing human affective states, containing data from 32 subjects who watched 40 one-minute long excerpts of music videos. Using the three-dimensional VAD model of Mehrabian and Russell [3], each of the six basic emotions has been defined as a combination of valence/arousal/dominance intervals [63]. Then, we classified them into two classes: 0—lack of emotion and 1—emotion by training and cross-validating using various machine learning and deep learning techniques, with and without feature selection. For anger, the highest classification accuracy has been obtained with Petrosian and Higuchi fractal dimensions, using SVM and no feature selection (98.02%). For joy, the highest classification accuracy has been achieved by LDA and SFS using raw EEG values (100%). For surprise—SVM and SFS with raw EEG values (96%), disgust—kNN with Petrosian values and no feature selection (95%), fear—kNN with raw EEG values and no feature selection (90.75%), and sadness—SVM with Higuchi fractal dimensions and no feature selection (90.8%). In the case of four emotions (anger, disgust, fear and sadness), the classification accuracies were higher without feature selection.

Our approach to emotion classification has applicability in the field of affective computing [64]. The domain includes all the techniques and methods used for the automatic recognition of emotions and their applications in healthcare, education, marketing, website personalization, recommendation systems, video games, and social media. Basically, human feelings are translated to the computers, which can understand and express them. The identification of the six basic emotions can be used for developing assistive robots, as the ones which detect and processes the affective states of children suffering from autism spectrum disorder [65], intelligent tutoring systems that use automatic emotion recognition to improve learning efficiency and adapt learning contents and interfaces in order to engage students [66], virtual reality games or immersive virtual environments that act as real therapists in anxiety disorder treatment [9,10], recommender systems which know the users' mood and adapt the recommended items accordingly [67,68], public sentiments analysis about different events, economic, or political decisions [69], and assistive technology [70–72].

Emotions play a central role in explainable artificial intelligence (AI), where there is so much need for human–AI interaction and human–AI interfaces [73]. As future research directions, we intend to classify the six basic emotions into three classes, namely negative, neutral, and positive and to develop emotion-based applications starting from the results presented in this paper, in the emerging field of explainable AI.

**Author Contributions:** Conceptualization, O.B., G.M., L.P.; methodology, O.B., G.M., L.P.; software, O.B.; validation, O.B., G.M., A.M., F.M., M.L.; investigation, O.B. and G.M.; resources, A.M. and M.L.; writing—original draft preparation, O.B., G.M., L.P.; writing—review and editing, O.B., G.M., L.P.; supervision, A.M., M.L. and F.M. All authors have read and agreed to the published version of the manuscript.

**Funding:** The work has been funded by the Operational Programme Human Capital of the Ministry of European Funds through the Financial Agreement 51675/09.07.2019, SMIS code 125125, UEFISCDI project 1/2018 and UPB CRC Research Grant 2017. This work has been funded in part through UEFISCDI, from EEA Grants 2014-2021, project number EEA-RO-NO-2018-0496.

**Conflicts of Interest:** The authors declare no conflict of interest.

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
