# Peer review of "Emotion Classification Based on Biophysical Signals and Machine Learning Techniques"

_symmetry, doi:10.3390/sym12010021_

Round 1

Reviewer 1 Report

Title of the Paper: Emotions classification based on biophysical signals and machine learning techniques

This paper reports on the applications of several machine learning techniques for classification of raw and processed biophysical signals (EEG etc.) with the ultimate goal of recognizing them with high performance and using that information to drive further decisions. The paper starts with a lengthy part about previous work dealing with mathematical expressions of emotions (as analyzed by different scientists), distilling them to distinct categories, so that different types of classifications can be applied on them.

1) Originality:  

The paper does not contain novel aspects. It must be decided by the editor-in-chief whether this justify the publication because it lacks novelty. The  goal is to make the right recommendation based on the correct identification of the feeling - this is really relevant - but there is no reporting on that metric. Does the performance increase affect the goal or not? This should be made clear by the authors! The feature selection does not comprise novelty as well. The machine learning techniques used are not specifically designed for the task. So either it must be completely revised or adapted as a survey paper - in either way this paper needs major revisions!

2) Related Work:

There are a lot of papers and references missing on the machine learning techniques themselves; here this reviewers gives only some ad-hoc hints to the emotion part, there are two more related relevant works that the authors can include:

page 17, valence/arousal ... line 451, here a pointer to related work for the interested reader would be important, e.g. [yy]

[yy]. Stickel, C. et al. Emotion detection: Application of the valence arousal space for rapid biological usability testing to enhance universal access. In Universal access in human-computer interaction. Addressing diversity, lecture notes in computer science, lncs 5614, Stephanidis, C., Ed. Springer: Berlin, Heidelberg, 2009; pp 615-624. 10.1007/978-3-642-02707-9_70

page 17, additonally to references [65], [66], [xx]

[xx]. Holzinger, A.; Bruschi, M.; Eder, W. On interactive data visualization of physiological low-cost-sensor data with focus on mental stress. In Multidisciplinary research and practice for information systems, springer lecture notes in computer science lncs 8127, Alfredo Cuzzocrea, C.K., Dimitris E. Simos, Edgar Weippl, Lida Xu, Ed. Springer: Heidelberg, Berlin, 2013; pp 469–480. 10.1007/978-3-642-40511-2_34

Generally the paper has potential, e.g. particularly for the upcoming field of what is now calle explainable ai and the for this necessary human-AI interaction and human-AI interfaces, here emotion will play a central role,

this can be mentioned in the last paragraphs in the future work section and at least one pointer should be given, e.g. [zz]

[zz]. Holzinger, A.; Kieseberg, P.; Weippl, E.; Tjoa, A.M. Current advances, trends and challenges of machine learning and knowledge extraction: From machine learning to explainable ai. In Springer lecture notes in computer science lncs 11015, Springer: Cham, 2018; pp 1-8. 10.1007/978-3-319-99740-7_1

3) Methodology:

The description of the emotions and their analysis as well as their relationships, which comprises the first half of the paper is extensive enough. But the machine learning methods, their architecture and parameters are not described and not justified sufficiently. The models are regarded as black boxes and the authors don’t make an effort to compare the strengths and weaknesses of each of them w.r.t. the others (see also the comment on explainable ai in section 2 above). For example, in the Discussion part they state that the “… most selected features ...” (I suppose they mean the most important feature) comes probably from the random forest? How did they find the most important feature in other models?

4) Results:

The results comprise long lists of accuracy and F1 scores. First of all, accuracy is not enough unless one has a balanced dataset and second of all the value of accuracy is almost all the time equal to the value of F1, which is very suspicious if one thinks the mathematical equation behind it. The fact that the details of the machine learning models (other than the neural networks) are not presented in an image (like for example how big is the depth of the tree) or in text. This is non reproducible research. 

5) Qualitative Evaluation:

The paper is well written and comprehensive. There were no understanding problems. The tables presenting the performance results are lengthy and could be compressed to distill the important information. Instead of presenting 2 pages of tables (with a lot of empty space), a decision tree structure or a visualization of the k-NN could make the research work more interesting and show the researchers insights necessary for further work.

Overall, I would recommend a major revision, but overall, despite the lack of novelty, I find the authors did a good job, and the paper can be interesting for the readers of the journal.

Author Response

Dear reviewer,

Thank you for your comments and suggestions aimed at improving the quality of our paper. Here are the punctual responses to your comments:

1) Originality:

The paper does not contain novel aspects. It must be decided by the editor-in-chief whether this justify the publication because it lacks novelty. The goal is to make the right recommendation based on the correct identification of the feeling - this is really relevant - but there is no reporting on that metric. Does the performance increase affect the goal or not? This should be made clear by the authors! The feature selection does not comprise novelty as well. The machine learning techniques used are not specifically designed for the task. So either it must be completely revised or adapted as a survey paper - in either way this paper needs major revisions!

In this paper we classify the six basic emotions from Ekman’s model into 2 classes (emotion and lack of emotion), by taking into account the emotion dimensions of valence-arousal-dominance from Mehrabian and Russell’s model. This approach is novel and original, as no one had before the idea of combining these two models and apply on them classical machine learning techniques and emerging deep learning algorithms, with and without feature selection (we used 3 different algorithms for feature selection). Usually, these two models are analyzed separately. In a paper published in April 2019 in the Sensors journal, we applied the same methods for classifying the emotion of fear, based on the recordings from the DEAP dataset. The current work is an extension and continuation of the paper published in Sensors:

“Fear Level Classification Based on Emotional Dimensions and Machine Learning Techniques” – Oana Balan, Gabriela Moise, Alin Moldoveanu, Marius Leordeanu, Florica Moldoveanu. Sensors 2019, 19(7), 1738; https://doi.org/10.3390/s19071738

Indeed, we have not used a database of our own, but this was because we wanted to test the classification algorithms on a reference multimodal dataset which has been largely used in various experiments for the analysis of human affective states, such as DEAP. Our goal was to see which algorithm works best for classifying emotions because our future research purpose is to design a system for treating phobias and anxiety disorders that automatically adapts the exposure scenarios based on the users’ physiological signals. A classification model will take as input the biophysical data and estimate the current emotional state. This model will be constructed based on data recorded in our experiments, but first, the idea was to see how well the state-of-the-art machine learning and deep learning techniques work on the data from DEAP.

Certainly, performance increase affects the goal of the research. A higher performance of the classification algorithms means that we can better divide a set of biophysical recordings into one of the two classes: emotion or lack of emotion, 0 or 1. In our future research, by applying the model, we can know if the user experiences fear or not, is angry or not, happy or not and we can adapt the exposure scenarios according to his emotional states. Of course, this applies not only to our studies, but also to others, in a large range of domains: game development, education, health, assistive technology and so on.

The feature selection algorithms are state-of-the-art techniques, widely used in thousands of studies, not only in emotion classification: PCA, Fisher score and SFS. We used three algorithms because we wanted to compare their results. They have been detailed in section 2.6, together with the deep learning and machine learning algorithms. Deep neural networks are new paradigms of classification, while the other techniques: SVM, LDA, kNN and RF are classical algorithms that provide very good classification accuracies.

2) Related Work:

There are a lot of papers and references missing on the machine learning techniques themselves; here this reviewers gives only some ad-hoc hints to the emotion part, there are two more related relevant works that the authors can include:

page 17, valence/arousal ... line 451, here a pointer to related work for the interested reader would be important, e.g. [yy]

[yy]. Stickel, C. et al. Emotion detection: Application of the valence arousal space for rapid biological usability testing to enhance universal access. In Universal access in human-computer interaction. Addressing diversity, lecture notes in computer science, lncs 5614, Stephanidis, C., Ed. Springer: Berlin, Heidelberg, 2009; pp 615-624. 10.1007/978-3-642-02707-9_70

page 17, additonally to references [65], [66], [xx]

[xx]. Holzinger, A.; Bruschi, M.; Eder, W. On interactive data visualization of physiological low-cost-sensor data with focus on mental stress. In Multidisciplinary research and practice for information systems, springer lecture notes in computer science lncs 8127, Alfredo Cuzzocrea, C.K., Dimitris E. Simos, Edgar Weippl, Lida Xu, Ed. Springer: Heidelberg, Berlin, 2013; pp 469–480. 10.1007/978-3-642-40511-2_34

Generally the paper has potential, e.g. particularly for the upcoming field of what is now calle explainable ai and the for this necessary human-AI interaction and human-AI interfaces, here emotion will play a central role,

this can be mentioned in the last paragraphs in the future work section and at least one pointer should be given, e.g. [zz]

[zz]. Holzinger, A.; Kieseberg, P.; Weippl, E.; Tjoa, A.M. Current advances, trends and challenges of machine learning and knowledge extraction: From machine learning to explainable ai. In Springer lecture notes in computer science lncs 11015, Springer: Cham, 2018; pp 1-8. 10.1007/978-3-319-99740-7_1

We have added the references and mentioned about Explainable AI in the Conclusions section.

3) Methodology:

The description of the emotions and their analysis as well as their relationships, which comprises the first half of the paper is extensive enough. But the machine learning methods, their architecture and parameters are not described and not justified sufficiently. The models are regarded as black boxes and the authors don’t make an effort to compare the strengths and weaknesses of each of them w.r.t. the others (see also the comment on explainable ai in section 2 above). For example, in the Discussion part they state that the “… most selected features ...” (I suppose they mean the most important feature) comes probably from the random forest? How did they find the most important feature in other models?

Please look at section 2.5. The entire text has been changed and we added a table containing a comparative study of the machine learning and deep learning techniques used for classifying emotions.

Also, in section 2.6 we have provided more details on the description, implementation and parameters of the deep learning and machine learning algorithms.

Indeed, we refer to the “most important features”, i.e. the features which are most relevant for classification. They have been provided by the feature selection algorithms - Fisher score and SFS algorithm (we added this to the text). 

4) Results:

The results comprise long lists of accuracy and F1 scores. First of all, accuracy is not enough unless one has a balanced dataset and second of all the value of accuracy is almost all the time equal to the value of F1, which is very suspicious if one thinks the mathematical equation behind it. The fact that the details of the machine learning models (other than the neural networks) are not presented in an image (like for example how big is the depth of the tree) or in text. This is non reproducible research.

The dataset used for training and testing, for each emotion, is balanced:

 The DEAP database contains 40 valence/arousal/dominance ratings for each of the 32 subjects. For the emotion of anger, there were 28 ratings in the database for Condition 1 – Anger and 239 ratings for the Condition 0 – No anger. In order to have a balanced distribution of responses for classification, we used 28 ratings for Condition 1 and 28 ratings for Condition 0, so we took the minimum between both. Every physiological recording had a duration of 60 s. Thus, in order to obtain a larger training database, we have divided the 60 s long recordings into 12 segments, each being 5 seconds long. Thus, for anger we obtained a training dataset of 672 entries that was fed to the classification algorithms. Table 5 presents, for each emotion, the number of entries for Conditions 0 and 1 and the total number of 5-s long segments that have been fed as input data to the classification algorithms.

Regarding the F1 score and accuracy, these are the results provided. We used the accuracy_score and f1_score methods from the scikit learn library. Here is the code:

kfold = KFold(n_splits=10, shuffle=True, random_state=seed)

model.fit(X,Y, epochs=100, batch_size=20, verbose=1)

y_pred = cross_val_predict(pipeline, X, Y, cv=kfold)

accuracy_score(y_pred.astype(int), Y.astype(int))

f1_score(Y, y_pred, average='weighted')*100

Regarding a graphical representation of the decision trees, we added Figure 3. Figure 3 presents the decision tree obtained for classifying Anger using RF with raw EEG data and peripheral features, without feature selection, with the highest F1 score and accuracy – 96.04%.

5) Qualitative Evaluation:

The paper is well written and comprehensive. There were no understanding problems. The tables presenting the performance results are lengthy and could be compressed to distill the important information. Instead of presenting 2 pages of tables (with a lot of empty space), a decision tree structure or a visualization of the k-NN could make the research work more interesting and show the researchers insights necessary for further work.

Thank you for your comments. We added Figure 3 with the decision tree. For kNN, the graphical representation in Python is available only if the input dataset has 2 features. Here is a representation of the kNN classification when the most relevant two features have been selected using the Fisher score, for classifying raw EEG data + peripheral data. The classification accuracy in this case is 79.7%. With the whole set of 20 most important features selected by the Fisher algorithm, the accuracy was 97.52%. Thus, more features add more accuracy in the classification.

Looking forward to hearing from you and we hope that you will find our paper suitable for publication.

Yours sincerely,

The authors

Reviewer 2 Report

The paper deals with a relevant research topic and the research method is correctly described.

However, the validation scheme is not correct. Indeed, each recording/rating of 60s is divided into 5s windows and mixed into the training/validation set. With the 10-fold cross validation scheme used with this set, each validation sample should have some corresponding training samples of the same recording in the training set. This should have led to overfitting that cannot be detected by the validation scheme. The authors should have used a leave one subject out (LOSO) to estimate the generalisation cabability of their models correctly. 

Author Response

Dear reviewer,

Thank you for your comments and suggestions aimed at improving the quality of our paper. Here are the punctual responses to your comments:

The paper deals with a relevant research topic and the research method is correctly described.

We are grateful for your appreciations. Indeed, this is a relevant research topic with a lot of potential in computer science, biology and psychology.

However, the validation scheme is not correct. Indeed, each recording/rating of 60s is divided into 5s windows and mixed into the training/validation set. With the 10-fold cross validation scheme used with this set, each validation sample should have some corresponding training samples of the same recording in the training set. This should have led to overfitting that cannot be detected by the validation scheme. The authors should have used a leave one subject out (LOSO) to estimate the generalization capability of their models correctly. 

We took into account your advice and in the short time we had for preparing the revised form of the manuscript (only 7 days), we run our artificial neural networks using the Leave-One-Out method with the implementation from the scikit learn library [1]. We observed that the computational cost was very high, because the training set was large: Anger – 672 entries, Joy – 2808, Surprise – 4824, Disgust – 1464, Fear – 1944, Sadness – 2136. The algorithm had to test on a sample and train on the rest of the samples. For all emotions, for all four DNN models, with four types of input, with and without feature selection, the time needed to run everything was really long. We observed that the accuracies were also high when cross-validating using the Leave-One-Out method and the hierarchy of results preserved, not affecting the classification ranking. The differences in accuracy between k-fold and Leave-One-Out were around 5-10%. This information has been added to the paper.

Cross-validation has been performed by using the k-fold method with k = 10 splits and the Leave-One-Out method, which takes each sample as test set and keeps the remaining samples in the training set. However, the Leave-One-Out method is more computationally demanding than k-fold. The model has been trained and cross-validated for 10 times and we calculated the average accuracy and F1 score across these 10 iterations. Each time, the input data has been shuffled before being divided into the training and test datasets.

The accuracies obtained using the Leave-One-Out method for cross-validation are with 5-10% lower, but the hierarchy of results is preserved, not affecting the classification ranking.

[1] https://scikit-learn.org/stable/modules/generated/sklearn.model_selection.LeaveOneOut.html

Looking forward to hearing from you and we hope that you will find our paper suitable for publication.

Yours sincerely,

The authors

Round 2

Reviewer 1 Report

The authors have addressed the reviewers comments in an adquate manner, so this reviewer does no longer argue against this paper.

Author Response

Dear reviewer,

Thank you for accepting our paper.

All the best, 

The authors

Reviewer 2 Report

Leave one out has the same problem with regard to k-fold.

What I suggested was to use leave one SUBJECT out (i.e. the whole data corresponding to one of the 15 participants being used as validation data for each learning, not only one sample). This should not have a high computational cost because there is only 15 participants (leading to 15 learing/validation only). Only this scheme verify the possiblity to generalize for new particpant. Another scheme could have been leave one video out ta verify the generalization for new videos.

Author Response

Dear reviewer,

Thank you for your comments and suggestions aimed at improving the quality of our paper. In the attached pdf you can find the complete responses to your comments and suggestions.
